# Defining the multifaceted roles of dental academics: A consensus approach using the Fuzzy Delphi Method

**Norasmatul Akma Ahmad**[1]*, **Noor Hayaty Abu Kasim**[1]*, **Sabri Musa**[2]*,
**Siti Adibah Othman**[2]*, **Zahra Naimie**[1]*

1 Department of Restorative Dentistry, Faculty of Dentistry, Universiti Malaya, Kuala Lumpur, Malaysia,
2 Department of Paediatric Dentistry and Orthodontics, Faculty of Dentistry, Universiti Malaya, Kuala Lumpur, Malaysia

* akma@um.edu.my (NAA); nhayaty@um.edu.my (NHAK); sabrim@um.edu.my (SM); sitiadibah@um.edu.my (SAO); z.naimie@gmail.com (ZN)

## Abstract:

Dental academics are the foundation of dental institutions. They are mostly dental specialists and scientists driven by a passion for teaching and intellectual challenges. Although most have not been trained or certified as educators, they are tasked with educating the next generation of dentists within their fields of expertise. Beyond teaching, dental academics juggle multifaceted roles as educators, researchers, administrators, and clinicians. The effort to balance these multiple roles may lead dental academics to face identity conflicts, loss of work satisfaction and ultimately burnout and attrition from the workforce. Given the limited literature on defining faculty competencies and roles in medical and dental education, and an ongoing retention issues, it is crucial to clearly delineate these roles to assist dental academics establish their academic identity and plan their careers. This study aims to systematically define the specific roles of dental academics using the Fuzzy Delphi Method (FDM). Engaging 27 experts in the first round and 23 in the second, consensus was reached on items derived from evidence-based literature to define these roles. Analysis yielded 17 accepted items representing the four main roles of dental academics. Each primary role encompassed a diverse range of job descriptions, precluding a single, definitive description. The application of FDM not only refines the understanding of academic roles but also contributes to establishing the identity of dental academics, aiding their adaptation to the multiple roles and supporting their career advancement.

## Introduction

Dental academics play a pivotal role in shaping the future of dental education, serving as the foundation of educational institutions while navigating a complex array of

**Data availability statement:** All relevant data are within the manuscript and its Supporting Information files.

**Funding:** The author(s) received no specific funding for this work.

**Competing interests:** The authors have declared that no competing interests exist.

responsibilities that extend well beyond traditional teaching. These professionals, often specialists in their respective fields, are charged with the critical task of preparing the next generation of dentists, despite many lacking formal pedagogical training [1]. The assumption that clinical expertise inherently qualifies them as educators can lead to a reliance on generic teaching principles, which may not adequately address the unique challenges of dental education [2,3], highlighting the need for professional development to enhance their effectiveness as teachers [4]. However, teaching is merely one facet of their multifaceted roles. In addition to instruction, dental academics are frequently involved in research, administrative duties, and clinical practice [5,6], which necessitates targeted professional development and institutional support.

Despite the growing recognition of the importance of faculty development programs tailored to the diverse skill sets required by dental educators [7], literature addressing the definition of faculty roles in dental education remains scarce. Scott (2003) identified four primary roles namely, teaching, research, clinical practice, and administration [8], while Hands (2006) proposed a tripartite classification encompassing clinical teachers, clinical scholars, and research-intensive scholars [9]. More recent classifications have reiterated these roles but emphasized the expectation for dental academics to balance multiple responsibilities due to their interconnected nature [5]. The struggle to manage diverse roles can lead to professional ambiguity, diminished job satisfaction, and ultimately contribute to faculty burnout and attrition within the dental workforce [10–12].

A clearer understanding and definition of dental academic roles are essential not only for enhancing faculty satisfaction and productivity but also for fostering career progression among dental academics. By delineating and reinforcing the distinct responsibilities associated with teaching, research, administration and clinical practices, institutions can provide targeted professional development opportunities that support faculty retention and facilitates the establishment of faculty's academic and professional identities. Academic identity encompasses self-definition and self-understanding shaped by social institutions and individual relationships [10] while professional identity is rooted in vocational self-concept and is regarded as an individual's role within a workplace [13].

A nuanced comprehension of academic roles, attributes, and competencies can guide dental academics in solidifying their academic and professional identities. Encouraging individuals to cultivate their skills across various roles will empower them to navigate their academic careers effectively [14]. Moreover, while identity is not the sole determinant of career success, it significantly influences motivation for learning and professional development goals [15], ultimately impacting recognition and rewards within academia [16]. A clear definition of dental academics' roles can foster a well-defined academic identity, which in turn enhances self-efficacy and job motivation, ultimately contributing to greater career success [17,18].

This study represents the first phase of a larger research project aimed at developing a comprehensive Dental Academic Career Pathway (DACP) model. It seeks to systematically define the specific roles of dental academics using the Fuzzy Delphi Method (FDM), forming a foundational step toward understanding the attributes,

competencies, and developmental needs within dental academia. Through expert consensus, this study offers essential role definitions that can support academic identity formation and inform institutional policy and professional development. While the questionnaire was constructed through literature mapping and expert input, the broader study adopts the Social Cognitive Career Theory (SCCT) [19] as its conceptual lens. SCCT will be applied in later phases to interpret how clearly defined academic roles may influence self-efficacy, outcome expectations, and motivation to pursue or sustain a career in dental academia.

## Materials and methods

### Ethical approval

Ethical approval for this study was granted by the Medical Ethics Committee, Faculty of Dentistry, Universiti Malaya (Ethics No: DF RD 2212/0070 [L]). Data collection was conducted from 31st January 2023–1st April 2023, with all participants providing informed consent through an online platform.

This study employed the Fuzzy Delphi Method (FDM), an advanced version of the traditional Delphi method that integrates fuzzy set theory to address the ambiguities often present in expert opinions [20]. The FDM involves a series of stages that must be followed for a study to be considered empirical, with seven specific steps outlined in FDM [21] (see Table 1).

### Fuzzy Delphi Method (FDM) questionnaire development

A FDM questionnaire was developed based on literature review. Databases such as ScienceDirect, IEEE Explore, PubMed and Web of Science were utilized to search for related literatures on the identified variables which in this study were the four main roles, teacher, researcher, administrator and clinician's definitions. In this questionnaire, each of the variables is referred to as a major element or a construct and each of the statements related to the variables are referred to as a sub-element or an item under construct (Fig 1).

For an item to be incorporated within a construct, it must be supported by the existing literature. Consequently, a literature mapping defining the various roles of dental academics was devised (Table 2).

The FDM questionnaire was designed using the SurveyMonkey platform (SurveyMonkey Inc., San Mateo, CA, USA; www.surveymonkey.com). It provided instructions for the expert panellists, detailing the purpose, duration, and objectives of the study, the investigator's contact information, and an overview of the Delphi process.

The questionnaire was pre-tested and subsequently piloted with 12 dental academics who were not part of the main expert panel. This pilot study was conducted prior to the first round of the Fuzzy Delphi Method to assess the reliability, clarity, and relevance of the questionnaire items. Amendments were made based on pre-test feedback to improve wording and structure. During the pilot phase, the internal consistency of the constructs was evaluated using Cronbach's alpha (α), standard deviation (SD), inter-item correlation (r), and corrected item-total correlation (CITC). Findings from the pilot test were used to refine item phrasing and ensure reliability before conducting the first round of the FDM. These results are reported under the 'Fuzzy Delphi Method Questionnaire Reliability' subheading.

### Expert panellists recruitment

Based on the inclusion and exclusion criteria (Table 3), a purposive sampling strategy was used to identify potential experts from three countries. Expert panellists were identified through institutional websites and publicly available curricula vitae, with selection based on their active involvement in multiple academic roles such as teaching, research, clinical practice, and administration.

Invitations were sent via email to 37 identified experts from 13 dental institutions and healthcare facilities, and those who consented to participate were subsequently provided with the Fuzzy Delphi Method (FDM) questionnaires through

**Table 1. The FDM steps.**

| STEPS | PROCEDURES |
|---|---|
| Step 1 | Preparation of Fuzzy Delphi questionnaires either through literature review, expert interviews, or focus groups discussion. |
| Step 2 | Identification of experts and the number of experts who will be appointed to assess the importance of evaluation criteria for elements, items, issues, variables, and other variables that will be measured using linguistic variables needs to be determined. |
| Step 3 | All linguistic variables are converted to triangular fuzzy numbers. |
| Step 4 | Conversion of the Likert scale data into a fuzzy scale and then, the threshold is calculated, i.e., (d) value based on this formula: $$d\left(\widetilde{m}, \widetilde{n}\right) = \sqrt{\frac{1}{3}\left[(m_1 - n_1)^2 + (m_2 - n_2)^2 + (m_3 - n_3)^2\right]}$$ This process involves converting all scales of linguistic variables to Fuzzy triangle numbering (Triangular Fuzzy Numbers). The values of $m_1$, $m_2$ and $m_3$ are used to represent the Triangular Fuzzy Number. The value of $m_1$ relates to the minimum value, $m_2$ to the most appropriate value, and $m_3$ to the maximum value |
| Step 5 | Determination of the first condition which is when the difference between the average and expert evaluation data is less than or equal to the threshold value, (d) = 0.2, all experts are said to have achieved an agreement. |
| Step 6 | Determination of the second condition, which is the percentage value for expert agreement must be more than or equal to 75.0%. |
| Step 7 | Defuzzification process which analyses data using an average of fuzzy numbers or an average response to identify the value of the fuzzy score and the rank and priority of each item, element, issue, and so on calculated based on this formula: $$A = \frac{1}{3}\left(m_1 + m_2 + m_3\right)$$ |

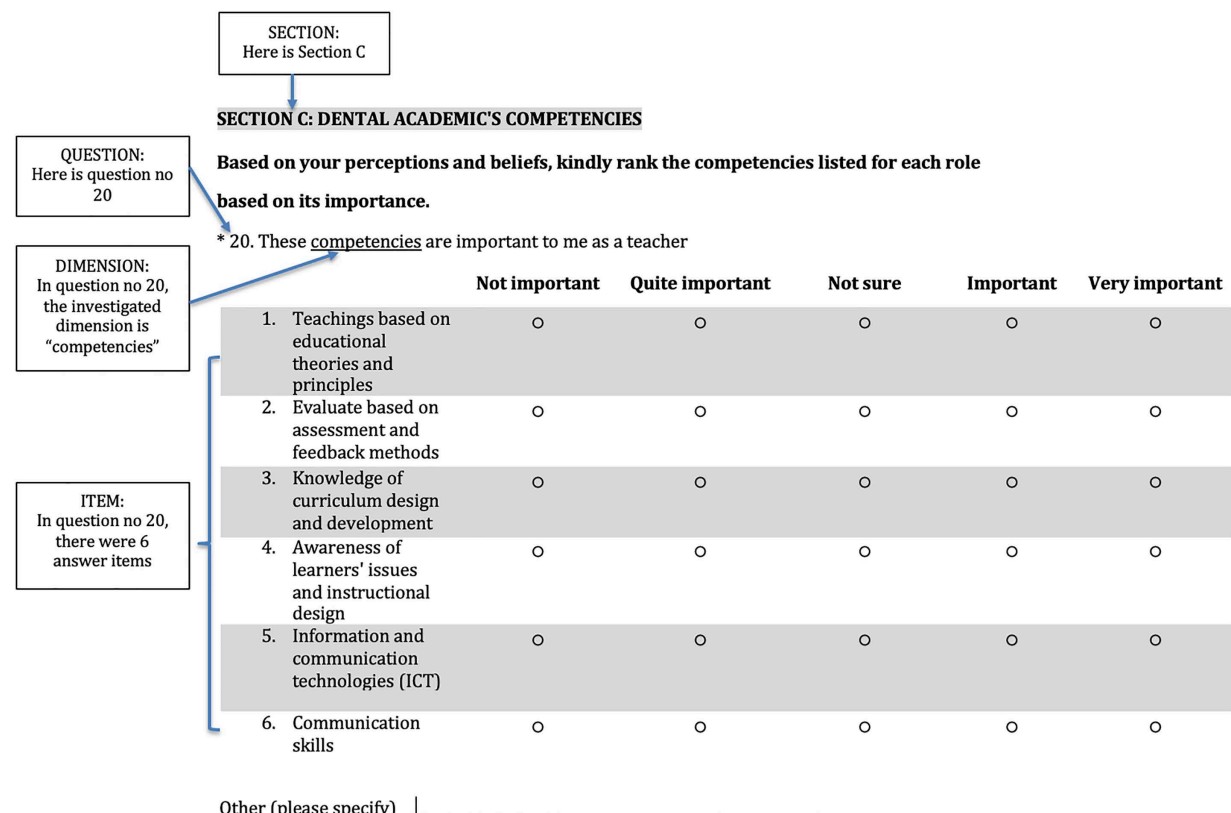

**Fig 1. An example of the major element and sub-elements in the FDM questionnaire.**

**Table 2. Literature mapping for the definition of different dental academics' roles.**

| MAJOR ELEMENT | SUB-ELEMENT | REFERENCES |
|---|---|---|
| Definition of a teacher | An educator whose main tasks are educating, guiding, teaching, directing, training, assessing, and evaluating students [22] | Saskianti et al.,2021 |
| | An individual who educates others in the academic settings with or without knowledge in dentistry [23] | Radford et al., 2015 |
| | An individual who is the expert of the field being taught in the academic settings [24] | Stenfors-Hayes et al, 2012 |
| | An individual who is trained and qualifiedwith Bachelor's or Master's degrees in the education field [25] | McConnell et al., 2021 |
| Definition of a researcher | An individual with specific research skills and specific research-related attributes [26] | Laidlaw et al., 2012 |
| | An active member of the Research Centre and/or primary investigator in at least one research project with human participants [27] | Cumyn et al., 2019 |
| | an individual who conducts research and provides direct patient care, although not necessarily at the same time or for the same organisation [28] | Hay-Smith et al., 2016 |
| | An individual whose primary tasks are to teach and conduct research at the university level [29] | Carter, 2019 |
| Definition of an administrator | An individual with knowledge, skills, and competencies regarding administration, managerial processes, legal and financial procedures related to an institution [30] | Fanoos & He, 2021 |
| | An individual who is appointed as programme directors, course coordinators, curriculum directors or clinical coordinators [31] | Weng & Tang, 2014 |
| | An individual who is expected to manage and lead the educational institution [25] | McConnell et al., 2021 |
| | An individual who retains a clinical role while also engaging in management related activities, such as strategic and collaborative work with health care managers and professionals [32] | Edmonstone, 2005 |
| Definition of a clinician | A healthcare professional that works in the healthcare setting [33] | Sherbino et al., 2014 |
| | An individual who delivers safe, efficient patient care, whist supervising and teaching students and trainees [34] | Cleland et al., 2018 |
| | An individual who is trained and highly skilled in clinical dentistry fields [35] | Kennedy and Hunt, 1999 |
| | An individual who serves as a clinical teacher in the academic settings [34] | Cleland et al., 2018 |

**Table 3. Inclusion and exclusion criteria for the FDM rounds.**

| Delphi panellist Inclusion Criteria | Delphi panellist Exclusion Criteria |
|---|---|
| Clinician and scientist who is currently serving or has previously served as an academic staff at a dental institution for more than 5 years. | Clinician and scientist who has not served as an academic staff at dental institution. |
| Lead administrative officer who is currently serving or has previously served as an administrative staff at dental institution for more than 5 years. | Clinician and scientist who is currently serving or has previously served as an academic staff at dental institution for less than 5 years. |
| A dental consultant who is currently serving or has previously served as a curriculum panellist at dental institution. | Lead administrative officer who is currently serving or has previously served as administrative staff at dental institution for less than 2 years. |
| Has experience as a leader or coordinator of a programme. | Dental consultant/ dentist who has not served as a curriculum panellist at dental institution. |
| | Has no experience as leader or coordinator of any educational programme. |

the SurveyMonkey platform. A summary of the demographic and professional characteristics of the expert panel is presented in Table 4.

### Fuzzy Delphi methods data analysis

All responses were compiled and tabulated in a Microsoft Excel worksheet and then transferred into the FDM Data Analysis template in Microsoft Excel (version 16.80). The first step in data analysis involved converting all linguistic variables into triangular fuzzy numbers (TFN). This process entails transforming the scales of the linguistic variables into TFN which

**Table 4. Demographic and Professional Background of Expert Panellists.**

| No. | Institutions | Country | Years in service | Designation |
|-----|-------------|---------|------------------|-------------|
| 1. | Public University | Malaysia | 33 | Professor |
| 2. | Public University | Malaysia | 32 | Professor |
| 3. | Public University | Malaysia | 15 | Assistant Professor |
| 4. | Public University | Malaysia | 15 | Assistant Professor |
| 5. | Public University | Malaysia | 9 | Associate Professor |
| 6. | Public University | Malaysia | 7 | Senior Lecturer |
| 7. | Public University | Malaysia | 15 | Senior Lecturer |
| 8. | Public University | Malaysia | 22 | Senior Lecturer |
| 9. | Public University | Malaysia | 20 | Associate Professor |
| 10. | Public University | Malaysia | 20 | Senior Lecturer |
| 11. | Public University | Malaysia | 21 | Senior Lecturer |
| 12. | Public University | Malaysia | 20 | Senior Lecturer |
| 13. | Public University | Malaysia | 24 | Senior Lecturer |
| 14. | Public University | Malaysia | 14 | Senior Lecturer |
| 15. | Public University | Malaysia | 24 | Senior Administrator |
| 16. | Public University | Indonesia | 16 | Senior Lecturer |
| 17. | Public University | Indonesia | 36 | Professor |
| 18. | Public University | Indonesia | 18 | Lecturer |
| 19. | Public University | Indonesia | 18 | Associate Professor |
| 20. | Private University | Indonesia | 30 | Professor |
| 21. | Private University | Malaysia | 35 | Professor |
| 22. | Private University | Malaysia | 30 | Associate Professor |
| 23. | Private University | Malaysia | 9 | Lecturer |
| 24. | Private University | Malaysia | 15 | Associate Professor |
| 25. | Private University | Turkey | 14 | Associate Professor |
| 26. | Ministry of Health | Malaysia | 26 | Clinical Consultant |
| 27. | Ministry of Health | Malaysia | 35 | Clinical Consultant |

were represented by the values m1, m2, and m3. Here, m1 corresponds to the minimum value, m2 to the most appropriate value, and m3 to the maximum value. In this study, the FDM questionnaire employed a 5-points Likert Scale and were converted into Fuzzy Scale based on their corresponding linguistic variable. [36–38]. Accordingly, each expert's response was mapped to its corresponding TFN within the Fuzzy Scale as presented in Table 5.

The Likert score given by each of the experts were inserted into the FDM analysis template and matched to their TFN and the average of m1, m2 and m3 were obtained for each of the items in the questionnaire. The difference between the

**Table 5. Conversion of Likert scale to Fuzzy Scale.**

| Likert scale | Linguistic Variables | Fuzzy Scales | | |
|--------------|---------------------|------|------|------|
| 1 | Strongly Disagree | 0.0 | 0.0 | 0.2 |
| 2 | Disagree | 0.0 | 0.2 | 0.4 |
| 3 | Neither Agree nor Disagree | 0.2 | 0.4 | 0.6 |
| 4 | Agree | 0.4 | 0.6 | 0.8 |
| 5 | Strongly Agree | 0.6 | 0.8 | 1.0 |

experts' evaluation data and the average value for each item, identified as the threshold value, 'd' was calculated using the formula as below:

$$d\left(\widetilde{m}, \widetilde{n}\right) = \sqrt{\frac{1}{3}\left[(m_1 - n_1)^2 + (m_2 - n_2)^2 + (m_3 - n_3)^2\right]}$$

Where: d = threshold value

m1 = average of minimum value, n1 = minimum value

m2 = average of plausible value, n2 = plausible value

m3 = average of maximum value, n3 = maximum value

This threshold "d" value is important in determining the levels of agreement among the panel experts upon the items. Following the establishment of the threshold value, the expert agreement percentage was computed for each item. The determination of group consensus was calculated by using the formula below [39]:

$$\frac{\text{Total of } d \leq 0.2 \times 100}{n \text{ of experts}}$$

It was specified that the group consensus percentage should surpass 75% for the subsequent steps to proceed. Upon reaching a collective consensus within the group, the defuzzification process was carried out using an average of fuzzy numbers. This calculation aims to determine the value of the fuzzy score (A), as well as the rank and priority of each item based on this formula:

$$A = \frac{1}{3}\left(m_1 + m_2 + m_3\right)$$

The item with the highest defuzzification or Fuzzy Score value (∝-cut value) will be ranked as the first item in the list. The fuzzy score value (A) must exceed the alpha cutting value of 0.5 (α-cut value > 0.5), for the item to be considered acceptable [40,41].

The consensus process in this study was conducted entirely through a quantitative approach. Each item in the questionnaire was analysed using the Fuzzy Delphi Method (FDM), and items were retained only if they met all three standard conditions: (1) the threshold value (d) reflecting the distance between experts' opinions was ≤ 0.2; (2) the percentage of agreement among panellists exceeded 75%; and (3) the fuzzy score (α-cut value) was greater than 0.5. These thresholds are consistent with established FDM practices and were applied to ensure that only items with strong consensus and perceived relevance were included. This rigorous and structured approach reduces subjectivity and enhances the reliability and validity of the findings.

### Delphi rounds

The number of rounds was determined based on the FDM analysis and the group consensus percentage. Once consensus exceeded 75% and data saturation was reached, no additional rounds were required. The same protocol and coordination were applied throughout all FDM rounds.

## Results

### Fuzzy Delphi Method questionnaire reliability

Based on the reliability analysis conducted during the pilot study, all four main constructs demonstrated acceptable to good internal consistency. The major element "definition as a clinician" achieved an alpha (α) of 0.760, while "definition as

a researcher" attained an alpha (α) of 0.775, both indicating acceptable reliability values. Similarly, "definition as an administrator" scored an alpha (α) of 0.806, and "definition as a teacher" scored an alpha (α) of 0.808, demonstrating a good level of reliability (Table 6). Given these findings, the questionnaire was deemed reliable and was retained without major modifications for use in the subsequent Delphi rounds. The finalised questionnaire consisted of 16 items representing various role definitions of dental academics.

## Expert panellists recruitment

The recruitment process for expert panellists yielded a favourable outcome, with approximately 73% of the invited experts (27 in total) agreeing to participate in the study. The initial panel consisted of 24 experienced dental academics, two senior clinical consultants affiliated with the Ministry of Health in Malaysia, and one senior administrator from a public institution. Following the first round, a second round of the Delphi method was conducted based on the analysis from the initial round. All 27 experts were invited to participate again via email; however, only 23 experts responded, resulting in an 85.2% response rate for this round. The composition of the second panel remained largely the same, with the exception of four experienced dental academics who withdrew.

## First FDM round data analysis

Based on the first FDM round (FDM 1), five out of the 16 response items did not comply with the FDM requirements (highlighted in Table 7). The FDM analysis revealed that item number 2, with a minimal 41% consensus among experts, a threshold value $d = 0.37$, and a Fuzzy score (A) $α = 0.447$, failed to meet the criteria of >75% expert consensus, $d ≤ 0.2$, and α-cut value >0.5. Similar findings were observed for item number 6. Regarding items number 4, 8, and 12, they did not meet two of the requirements. The three items scored $d > 0.2$ and group consensus <75% (Table 7). Consequently, all five items were excluded from the list while the other 11 items were kept for the second FDM round.

A mapping system was devised to discern the items in the FDM questionnaire according to their respective numbers. This mapping proved valuable for scrutinizing the items in light of the first FDM analysis and for the formulation of the second FDM questionnaire. The non-compliant items were highlighted in Table 8.

## Second FDM round data analysis

In the second round of the Delphi method (FDM 2), both quantitative and qualitative findings from the initial round (FDM 1) were considered. The second FDM questionnaire was developed based on the 11 items approved in FDM 1, supplemented by an additional nine items generated from expert feedback. This resulted in a comprehensive questionnaire

**Table 6. Internal consistency of the major elements in the FDM questionnaire.**

| Constructs | Cronbach's alpha,α (α>0.7) | Standard deviation, S (S>0) | Inter-item correlation, r (0.3>r<0.9) | Minimum Corrected Item-Total Correlation (CITC) (CITC >0.3) |
|---|---|---|---|---|
| Definition as a teacher (4 items) | 0.808 | 0.45227–0.51493 | 0.408–0.837 | 0.422 |
| Definition as a researcher (4 items) | 0.775 | 0.45227–0.51493 | 0.393- 0.683 | 0.446 |
| Definition as an administrator (4 items) | 0.806 | 0.49237–0.57735 | 0.426-0.746 | 0.510 |
| Definition as a clinician (4 items) | 0.760 | 0.49237–0.75378 | 0.490- 0.816 | 0.404 |

**Table 7. First FDM analysis with the excluded items highlighted.**

| | Triangular Fuzzy Number | | Defuzzification Process | | | | | | | |
|---|---|---|---|---|---|---|---|---|---|---|
| Items | Threshold value, d | Group consensus, % | m1 | m2 | m3 | Fuzzy Score (A) | Expert Consensus | ∝-cut value | Ranking based on definition |
| 1 | 0.10 | 96% | 0.56 | 0.76 | 0.96 | 0.76296 | ACCEPT | 0.763 | 2 |
| 2 | 0.37 | 41% | 0.27 | 0.44 | 0.64 | 0.44691 | REJECT | 0.447 | 16 |
| 3 | 0.15 | 96% | 0.53 | 0.73 | 0.93 | 0.72593 | ACCEPT | 0.726 | 3 |
| 4 | 0.31 | 52% | 0.31 | 0.50 | 0.70 | 0.50617 | REJECT | 0.506 | 14 |
| 5 | 0.18 | 85% | 0.45 | 0.65 | 0.85 | 0.65185 | ACCEPT | 0.652 | 9 |
| 6 | 0.28 | 59% | 0.28 | 0.47 | 0.67 | 0.47654 | REJECT | 0.477 | 15 |
| 7 | 0.13 | 89% | 0.41 | 0.61 | 0.81 | 0.614815 | ACCEPT | 0.615 | 11 |
| 8 | 0.26 | 56% | 0.36 | 0.56 | 0.76 | 0.562963 | REJECT | 0.563 | 13 |
| 9 | 0.17 | 93% | 0.47 | 0.67 | 0.87 | 0.666667 | ACCEPT | 0.667 | 7 |
| 10 | 0.17 | 93% | 0.48 | 0.68 | 0.88 | 0.681481 | ACCEPT | 0.681 | 6 |
| 11 | 0.18 | 93% | 0.47 | 0.67 | 0.87 | 0.666667 | ACCEPT | 0.667 | 8 |
| 12 | 0.20 | 44% | 0.40 | 0.60 | 0.80 | 0.600000 | REJECT | 0.600 | 12 |
| 13 | 0.20 | 81% | 0.44 | 0.64 | 0.84 | 0.644444 | ACCEPT | 0.644 | 10 |
| 14 | 0.06 | 100% | 0.58 | 0.78 | 0.98 | 0.777778 | ACCEPT | 0.778 | 1 |
| 15 | 0.18 | 93% | 0.51 | 0.71 | 0.91 | 0.711111 | ACCEPT | 0.711 | 4 |
| 16 | 0.19 | 93% | 0.50 | 0.70 | 0.90 | 0.703704 | ACCEPT | 0.704 | 5 |

**Table 8. First FDM analysis mapping with the excluded items highlighted.**

| ROLES | ITEMS NUMBER | DEFINITION ITEMS | FDM RANKING |
|---|---|---|---|
| Teacher | 1 | An educator whose main tasks are educating, guiding, mentoring, teaching, directing, training, assessing, and evaluating students on the knowledge of dentistry and other related sciences | 2 |
| | 2 | A professional with a bachelors/master's qualification in dentistry and a mandatory basic training in teaching | 16 |
| | 3 | An individual who is the expert of the field being taught in the academic settings. | 3 |
| | 4 | An individual who is trained and qualified with Bachelor's or Master's degrees in the education field | 14 |
| Researcher | 5 | An individual with specific research skills and specific research-related attributes. | 9 |
| | 6 | An active member of the Research Centre and/or primary investigator in at least one research project with human participants | 15 |
| | 7 | An individual who conducts research and provides direct patient care, although not necessarily at the same time or for the same organisation | 11 |
| | 8 | An individual whose primary tasks are to teach and conduct research at the university level. | 13 |
| Administrator | 9 | An individual with knowledge, skills, and competencies regarding administration, managerial processes, legal and financial procedures related to an institution. | 7 |
| | 10 | An individual who is appointed as programme directors, course coordinators, curriculum directors or clinical coordinators | 6 |
| | 11 | An individual who is expected to manage and lead the educational institution. | 8 |
| | 12 | An individual who maintains a clinical role while engaging in management- related activities, such as strategic and collaborative work with managers and health care professionals. | 12 |
| Clinician | 13 | A healthcare professional that works in the healthcare setting | 10 |
| | 14 | An individual who delivers safe, efficient patient care, whilst supervising and teaching students and trainees | 1 |
| | 15 | An individual who serves as a clinical teacher in the academic settings. | 4 |
| | 16 | An individual who is trained and highly skilled in clinical dentistry fields. | 5 |

comprising a total of 20 items, with 23 experts remaining as panellists. The FDM analysis revealed that three items did not meet the established criteria for retention (see Table 9).

Notably, item number 4 achieved only a 39% group consensus, with a threshold value of d = 0.31, which exceeds the acceptable limit. Similarly, item number 20 also recorded a 39% group consensus and a d value of 0.260, both of which were above the acceptable threshold, leading to their exclusion from further consideration. While item number 9 satisfied two of the FDM criteria, it fell short in terms of group consensus, achieving only 65%, which is below the acceptable value of 75%. Consequently, this item was also excluded from the final list. The item mapping process proved instrumental in identifying these items that did not meet retention criteria, as detailed in Table 10.

In the second round of the Delphi method (FDM 2), consensus was achieved on 17 items among the 23 participating experts, encompassing the four primary roles of dental academics The absence of further input indicated that data saturation had been attained. The accepted items are systematically ranked and presented in Fig 2, which tabulates the final definitions for each of the four main dental academic roles, as teacher, researcher, administrator, and clinician as derived through the FDM consensus process.

## Discussion

Dental academics are required to manage a wide range of responsibilities including teaching, research, clinical duties, and administration which contributes to a demanding work environment. This complexity can lead to challenges such as academic identity crisis [42], reduced job satisfaction, and hindered career progression. Regardless of the factors that motivate clinicians to pursue academic careers, whether it is limited awareness and unclear pathways that may discourage entry [43], or the appeal of intellectual engagement and job security that may encourage it [44] defining the core roles of dental academics remains essential. Clear role definitions are important to support academic identity development,

**Table 9. Second FDM analysis with the excluded items highlighted.**

| Items | Threshold value, d | Group consensus, % | m1 | m2 | m3 | Fuzzy Score (A) | Expert Consensus | ∝-cut value | Ranking based on definition |
|---|---|---|---|---|---|---|---|---|---|
| 1 | 0.09 | 100% | 0.57 | 0.77 | 0.97 | 0.76522 | ACCEPTED | 0.77 | 1 |
| 2 | 0.13 | 100% | 0.54 | 0.74 | 0.94 | 0.73913 | ACCEPTED | 0.74 | 4 |
| 3 | 0.20 | 78% | 0.46 | 0.66 | 0.86 | 0.66087 | ACCEPTED | 0.66 | 10 |
| 4 | 0.31 | 39% | 0.37 | 0.57 | 0.77 | 0.56522 | REJECTED | 0.57 | 18 |
| 5 | 0.20 | 87% | 0.46 | 0.66 | 0.86 | 0.66087 | ACCEPTED | 0.66 | 10 |
| 6 | 0.18 | 96% | 0.48 | 0.68 | 0.88 | 0.67826 | ACCEPTED | 0.68 | 8 |
| 7 | 0.17 | 96% | 0.47 | 0.67 | 0.87 | 0.66957 | ACCEPTED | 0.67 | 9 |
| 8 | 0.20 | 83% | 0.43 | 0.63 | 0.83 | 0.63478 | ACCEPTED | 0.63 | 13 |
| 9 | 0.18 | 65% | 0.39 | 0.59 | 0.79 | 0.59130 | REJECTED | 0.59 | 16 |
| 10 | 0.20 | 83% | 0.43 | 0.63 | 0.83 | 0.62899 | ACCEPTED | 0.63 | 14 |
| 11 | 0.15 | 91% | 0.43 | 0.63 | 0.83 | 0.63478 | ACCEPTED | 0.63 | 12 |
| 12 | 0.15 | 100% | 0.50 | 0.70 | 0.90 | 0.69565 | ACCEPTED | 0.70 | 6 |
| 13 | 0.17 | 91% | 0.44 | 0.64 | 0.84 | 0.64348 | ACCEPTED | 0.64 | 11 |
| 14 | 0.16 | 87% | 0.43 | 0.63 | 0.83 | 0.62609 | ACCEPTED | 0.63 | 15 |
| 15 | 0.15 | 91% | 0.44 | 0.64 | 0.84 | 0.64348 | ACCEPTED | 0.64 | 11 |
| 16 | 0.20 | 91% | 0.49 | 0.69 | 0.89 | 0.68696 | ACCEPTED | 0.69 | 7 |
| 17 | 0.10 | 100% | 0.56 | 0.76 | 0.96 | 0.75652 | ACCEPTED | 0.76 | 2 |
| 18 | 0.17 | 91% | 0.51 | 0.71 | 0.91 | 0.71304 | ACCEPTED | 0.71 | 5 |
| 19 | 0.13 | 100% | 0.54 | 0.74 | 0.94 | 0.73913 | ACCEPTED | 0.74 | 3 |
| 20 | 0.26 | 39% | 0.39 | 0.59 | 0.79 | 0.59130 | REJECTED | 0.59 | 17 |

**Table 10. Second FDM analysis mapping with the excluded items highlighted.**

| Roles | Items number | Definition | FDM ranking |
|---|---|---|---|
| Teacher | 1 | An educator whose main tasks are educating, guiding, mentoring, teaching, directing, training, assessing and evaluating students in the knowledge of dentistry and other related fields. | 1 |
| | 2 | An individual who is the expert of the field being taught in the academic settings and practices evidence-based dentistry, and evidence-based education. | 4 |
| | 3 | An individual who is trained and qualified with minimum qualification of a master's degree in dentistry and/or another related field. | 10 |
| | 4 | An individual who is trained and qualified in their scope of teaching with reasonable academic training and have a general overview of dentistry. | 18 |
| | 5 | An individual who has adequate clinical experience and is able to guide the students using teaching methods that are updated and relevant. | 10 |
| Researcher | 6 | An individual with specific research skills and specific research-related attributes whose secondary tasks are to conduct, teach/guide research projects for the improvement of patients, society, organisation and betterment of mankind. | 8 |
| | 7 | An individual who conducts research and develop new techniques or new knowledge that has the potential to change future dental practices. | 9 |
| | 8 | An individual who has the ability and passion to conduct research in their area of interest and provides direct patient care, although not necessarily at the same time or for the same organisation. | 13 |
| | 9 | An individual whose primary tasks are to teach and conduct research related to learning materials, to develop the sciences for educational purposes and not limited to the university or/and community level. | 16 |
| | 10 | An individual with basic research skills and knowledge such that they could provide supervision to the students. | 14 |
| Administrator | 11 | An individual with knowledge, leadership skills, and competencies regarding administration, managerial processes, legal and financial procedures related to an institution. | 12 |
| | 12 | An individual who may be appointed as programme directors, course coordinators, curriculum directors or clinical coordinators and helps to develop curriculum and manages academic and student programme. | 6 |
| | 13 | An individual who is expected to manage and lead the educational institution which involves strategic and collaborative activities. | 11 |
| | 14 | An individual who is appointed to carry out managerial or administrative work within the academic institution to facilitate the running and operations of academic, clinical, research and collaborative work so that other people (academicians/support staffs, students) can do their job well. | 15 |
| | 15 | An individual who is responsible for the dental program or parts of the program, talent management and overall operations of the dental school at the level suited for the appointment. | 11 |
| Clinician | 16 | A healthcare professional who works in the healthcare setting and involves directly in patient management, whether in clinic, hospital wards or operating theatres and actively provide treatment for patients to improve their wellbeing. | 7 |
| | 17 | An individual who serves as a clinical teacher in academic settings, and delivers safe, efficient patient care, whilst supervising and teaching students and trainees | 2 |
| | 18 | A specialist who is trained and highly skilled in clinical dentistry fields. | 5 |
| | 19 | An individual who is responsible in educating dental students, dental professionals, providing direct patient care and involved in its relationship with other disciplines which may be non-dental for the holistic care of patients. | 3 |
| | 20 | An individual who is trained in basic dental skills such that they can impart their skills to the students and deliver optimum health care standard themselves with passion for continuous education/ training. | 17 |

guide career progression, and inform institutional policies across diverse educational settings. Furthermore, the limited literature outlining faculty competencies and role expectations in medical and dental education [45] highlights a significant gap that this study seeks to address.

| DEFINITION | | | |
|---|---|---|---|
| **TEACHER** | **RESEARCHER** | **ADMINISTRATOR** | **CLINICIAN** |
| an educator whose main tasks are **educating, guiding, mentoring, teaching, directing, training, assessing and evaluating students** in the knowledge of dentistry and other related fields. | an individual with specific research skills and specific research-related attributes whose **secondary tasks are to conduct, teach/guide research projects** for the improvement of patients, society, organization and betterment of mankind. | an individual **who may be appointed as programme directors, course coordinators, curriculum directors or clinical coordinators** and helps to develop **curriculum** and **manages academic and student programme**. | an individual who **serves as a clinical teacher in academic settings**, and **delivers safe, efficient patient care**, whilst **supervising and teaching students and trainees** |
| an individual **who is the expert of the field being taught** in the academic settings and practices evidence-based dentistry, and evidence-based education. | an individual **who conducts research and develop new techniques or new knowledge** that has the potential to change future dental practices. | an individual who is **expected to manage and lead the educational institution** which involves strategic and collaborative activities. | an individual who is **responsible in educating dental students, dental professionals**, providing direct patient care and involved in its relationship with other disciplines which may be non-dental for the holistic care of patients. |
| an individual **who is trained and qualified with minimum qualification of a Master's degree** in Dentistry and/or another related field. | an individual **who has the ability and passion to conduct research** in their area of interest and provides direct patient care, although not necessarily at the same time or for the same organisation. | an individual who is **responsible for the dental program or parts of the program, talent management** and **overall operations of the dental school** at the level suited for the appointment. | a specialist who is **trained and highly skilled in clinical dentistry fields**. |
| an individual **who has adequate clinical experience and is able to guide the students** using teaching methods that are updated and relevant. | an individual with **basic research skills and knowledge such that they could provide supervision** to the students | an individual with **knowledge, leadership skills, and competencies regarding administration, managerial processes, legal and financial procedures** related to an institution. | a **healthcare professional** who works in the healthcare setting and **involves directly in patient management, whether in clinic, hospital wards or operating theatres** and **actively provide treatment for patients to improve their wellbeing**. |
| | | An individual who is appointed to **carry out managerial or administrative work** within the academic institution to **facilitate the running and operations of academic, clinical, research and collaborative work** so that other people (academicians/support staffs, students) can do their job well. | |

**Fig 2. Final accepted definitions of dental academic roles.** This figure presents the systematically ranked and categorised definitions of the four main roles as teacher, researcher, administrator, and clinician based on expert consensus using the Fuzzy Delphi Method.

In the attempt to understand and defined dental academics roles, the Fuzzy Delphi Method was employed. This approach was proposed by Murray et al. (1985) to enhance consensus-building among experts by incorporating fuzzy logic, which allows for more nuanced interpretations of their responses [46]. By using triangular fuzzy numbers, FDM enables a more precise representation of expert consensus, facilitating clearer decision-making and enhancing the reliability of the findings by improving the accuracy of the analysis [36]. This methodology is particularly valuable in contexts where expert opinions are critical, as it allows for a comprehensive understanding of complex issues while minimizing misinterpretations.

The careful selection of experts for the FDM rounds is crucial because the Delphi method does not require random statistical sampling; instead, participants must be chosen from qualified experts. While non-random sampling can introduce bias, the use of purposive sampling with strict inclusion and exclusion criteria in this study ensures sufficient diversity and transparency, mitigating this bias [46]. Furthermore, recruiting experts included clinicians and scientists with over 5 years of experience in academia, qualifies someone as an expert [47]. The heterogeneous panel of experts added in capturing a broad range of opinions. Involving experts and stakeholders from different geographical regions and areas of expertise enhanced the scientific robustness of the findings [48,49].

The definitions of dental academics, as derived from expert consensus, reveal a nuanced understanding of their roles and responsibilities within the educational landscape. The application of the FDM has illuminated the multifaceted nature of these roles, indicating that they cannot be encapsulated by a singular definition. Dental academics serve primarily as teachers, guiding and mentoring students while assessing their progress, all underpinned by their expertise in the field.

In addition to their teaching responsibilities, they are also researchers, equipped with specialized skills that enable them to conduct research and foster innovation in dental techniques and knowledge. Furthermore, as administrators, dental academics take on leadership roles such as program directors and curriculum coordinators, where they are instrumental in shaping educational policies and managing institutional operations. Lastly, in their capacity as clinicians, they provide critical patient care while simultaneously educating the next generation of dental professionals. This comprehensive framework underscores the complexity of dental academia and highlights the importance of recognizing the diverse competencies required for effective practice in each role. Such insights are essential for developing targeted faculty development programs that enhance the professional identities and effectiveness of dental educators.

The findings from this study not only enhance our understanding of these roles but also underscore the importance of creating supportive institutional structures that promote career satisfaction and growth among dental faculty [50]. By clarifying these roles, dental schools can better tailor faculty development programs to address the diverse competencies required for success in academia.

Through these definitions, individual dental academic may also reflect who they are or who they have been. This is crucial for initiating career planning and managing career transitions [51,52]. It provides a foundational platform for dental academics to align their current or past identities with their future career goals [53]. In addition, these definitions provide a framework to help dental academics adapt to their multifaceted responsibilities and to support their professional advancement [44]. Furthermore, these findings may act as a self-reflective support tool to establish their academic identity, emphasize the need for dental institutions to invest in targeted faculty development programs that address the broad competencies required for success in academia [12].

## Conclusion

A single definition was not discernible for each of the roles, given that each primary role encompassed a diverse range of job descriptions. Based on expert consensus, the following definitions outline the various roles within dental academia:

Dental academic as a teacher can be defined as an educator whose main tasks are educating, guiding, mentoring, teaching, directing, training, assessing and evaluating students in the knowledge of dentistry and other related fields or an individual who is the expert of the field being taught in the academic settings and practices evidence-based dentistry, and evidence-based education.

Dental academic as a researcher can be defined as an individual with specific research skills and specific research-related attributes whose secondary tasks are to conduct, teach/guide research projects for the improvement of patients, society, organization and betterment of mankind or an individual who conducts research and develop new techniques or new knowledge that has the potential to change future dental practices.

Dental academic as an administrator can be defined as an individual who may be appointed as programme directors, course coordinators, curriculum directors or clinical coordinators and helps to develop curriculum and manages academic and student programme or an individual who is expected to manage and lead the educational institution which involves strategic and collaborative activities.

Dental academic as a clinician can be defined as an individual who serves as a clinical teacher in academic settings, and delivers safe, efficient patient care, whilst supervising and teaching students and trainees or an individual who is responsible in educating dental students, dental professionals, providing direct patient care and involved in its relationship with other disciplines which may be non-dental for the holistic care of patients.

## Supporting information

**S1 Data. Sample Data for FDM.**
(XLSX)

## Acknowledgments

This study was self-funded by the first author as part of her doctoral research. We extend our gratitude to all the expert panellists for their invaluable support and contributions to this research.

## Author contributions

**Conceptualization:** Norasmatul Akma Ahmad.

**Data curation:** Norasmatul Akma Ahmad.

**Formal analysis:** Norasmatul Akma Ahmad.

**Investigation:** Norasmatul Akma Ahmad.

**Methodology:** Norasmatul Akma Ahmad, Zahra Naimie.

**Project administration:** Norasmatul Akma Ahmad, Noor Hayaty Abu Kasim, Sabri Musa.

**Resources:** Norasmatul Akma Ahmad.

**Supervision:** Noor Hayaty Abu Kasim, Sabri Musa, Siti Adibah Othman.

**Validation:** Norasmatul Akma Ahmad.

**Writing – original draft:** Norasmatul Akma Ahmad.

**Writing – review & editing:** Norasmatul Akma Ahmad, Noor Hayaty Abu Kasim, Sabri Musa, Siti Adibah Othman, Zahra Naimie.

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
