## [Decision Letter · Decision Letter 0]

30 May 2025

Dear Dr. Ahmad,

Thank you for submitting your manuscript to PLOS ONE. After careful consideration, we feel that it has merit but does not fully meet PLOS ONE’s publication criteria as it currently stands. Therefore, we invite you to submit a revised version of the manuscript that addresses the points raised during the review process.

We look forward to receiving your revised manuscript.

Kind regards,

Bhupendra Gopalbhai Prajapati, Ph.D., M.Pharm

Academic Editor

PLOS ONE

Journal Requirements:

Please confirm at this time whether or not your submission contains all raw data required to replicate the results of your study. Authors must share the “minimal data set” for their submission. PLOS defines the minimal data set to consist of the data required to replicate all study findings reported in the article, as well as related metadata and methods (https://journals.plos.org/plosone/s/data-availability#loc-minimal-data-set-definition ).

Reviewers' comments:

Reviewer's Responses to Questions

**Comments to the Author**

1. Is the manuscript technically sound, and do the data support the conclusions?

Reviewer #1: Yes

Reviewer #2: Yes

2. Has the statistical analysis been performed appropriately and rigorously?

Reviewer #1: I Don't Know

Reviewer #2: Yes

3. Have the authors made all data underlying the findings in their manuscript fully available?

Reviewer #1: No

Reviewer #2: Yes

4. Is the manuscript presented in an intelligible fashion and written in standard English?

Reviewer #1: Yes

Reviewer #2: Yes

Reviewer #1: This research addresses issues pertinent to dental academics in as far as their roles are concerned. Although the current literature was cited, it does not seem to reflect the global context. Some articles e.g. Ref 42 interpretation may need to be revised. In the introduction the authors mention the struggle to manage diverse roles leading to professional ambiguity, this leaves one with the question is it the struggle to manage the diverse roles or is that the roles are not well defined? It not clear how the diverse role bring about professional ambiguity, this needs to be clarified. In the introduction not much is said about the different roles in question. A brief insight or discussion on what the different roles being studied entails was not well presented in the introduction. It was not well demonstraded how the different roles plays out in the dental academics. This would have helped with understandning how the questionnaires were formulated.

The he study 42 cited with reference to career pathways in my understanding was more on lack of awareness about the career pathways in academia. Not sure the interpretation is corrects.

You can check out this article for some perspective of what’s happening in other countries.

Mostert VC. Reasons why South African dentists chose a career in Dentistry, and later opted to enter an academic environment. S. Afr. dent. j. [Internet]. 2018 Apr [cited 2025 Jan 30] ; 73( 3 ): 141-145. Available from: http://www.scielo.org.za/scielo.php?script=sci_arttext&pid=S0011-85162018000300006&lng=en.

The study design is appropriate, and has academic merit as it attempts to address an important topic that could address issues in dental education requiring some extent of committed and support to dental academics among key decision makers in dental education.

Details of methods and analysis applied in this study can be improved. It is not clear how the experts panelists where identified? The results of the pilot study for the Fuzzy Delphi Method was not anlyzed and published. This results should have been published or presented as it would have provided valuable insights into the effectiveness of the questionnaire and justification for refining before conducting first round of FDM. This could also assist those wishing to repeat similar study. Result of the pilot study could contribute to the understandning of FDM application and potential challenges.

There are no clearly defined objectives; the Fuzzy Delphi method is designed to gather expert consensus on a specific topic, so the research needs to have a clear goal in mind to guide the selection of experts, the questions asked, and the interpretation of results. These objectives should be aligned with the current knowledge and concerns within the field of study.

There are a number of issues with the methods and analysis that need to be clarified/addressed; furthermore, some of the conclusions overreach the data collected, while other important results are given less emphasis that they warrant. How where the experts identified, and how was their annonimity maintained. It is not clear how the individual expert opinions were aggregated and adjusted towards a consensus. This should be reported.

It is overall a valuable study, in institutions where the differen roles are not clearly defined.

Reviewer #2: This work aims to systematically define the specific roles of dental academics using the Fuzzy Delphi Method (FDM). Engaging 27 experts in the first round and 23 in the second, consensus was reached on items derived from evidence- based literature to define these roles. Analysis yielded 17 accepted items representing the four main roles of dental academics. Each primary role encompassed a diverse range of job descriptions, precluding a single, definitive description. The application of FDM not only refines the understanding of academic roles but also contributes to establishing the identity of dental academics, aiding their adaptation to the multiple roles and supporting their career advancement. Paper is well written and structure. My quires and suggestions are given as follows:

1) What exact criteria or decision rules were used in selecting and modifying the items between FDM rounds? Please provide examples of how feedback was used to generate the 9 new items in Round 2.

2) Can the authors clarify how subjectivity in assigning fuzzy numbers was minimized? Was there any calibration or standardization step among raters?

3) More information is needed on the pre-testing phase of the FDM questionnaire (e.g., how many items were revised or removed, types of issues identified, specific actions taken).

4) While Table 2 maps references to role definitions, not all added or retained items in Round 2 seem clearly traceable to literature. Could the authors provide supporting references for each final item?

5) Can the authors position their findings more clearly within established academic/professional identity frameworks? For instance, how do their role definitions align with or differ from Boyer's model or Wenger's Communities of Practice?

6) The manuscript asserts that roles are distinct, yet many items blend responsibilities (e.g., clinician and teacher). Could the authors more clearly distinguish or justify this overlap in terms of role theory?

7) The conclusion mentions support for academic identity and career planning. Could the authors provide a conceptual model or visual framework showing how these roles integrate into dental academic career stages?

8) Figure 2 is referenced but not adequately labeled or described. Can the authors ensure all figures and tables are self-explanatory and consistently formatted?

9) Several passages could benefit from grammatical tightening and clarity (e.g., “The strive to balance” → “The effort to balance”; “has experiences” → “has experience”). Consider a language polish throughout.

**Do you want your identity to be public for this peer review?** For information about this choice, including consent withdrawal, please see our Privacy Policy

Reviewer #1: **Yes: ** Shumani Charlotte Manenzhe

Reviewer #2: **Yes: ** Arvind Kumar Prajapati

---

## [Author Response · Author response to Decision Letter 1]

24 Jun 2025

Defining the Multifaceted Roles of Dental Academics: A Consensus Approach Using the Fuzzy Delphi Method

Reviewers’ Comments

Reviewer 1:

1) This research addresses issues pertinent to dental academics in as far as their roles are concerned. Although the current literature was cited, it does not seem to reflect the global context. Some articles e.g. Ref 42 interpretation may need to be revised.

The study 42 cited with reference to career pathways in my understanding was more on lack of awareness about the career pathways in academia. Not sure the interpretation is correct.

You can check out this article for some perspective of what’s happening in other countries.

Mostert VC. Reasons why South African dentists chose a career in Dentistry, and later opted to enter an academic environment. S. Afr. dent. j. [Internet]. 2018 Apr [cited 2025 Jan 30] ; 73( 3 ): 141-145. Available from: http://www.scielo.org.za/scielo.php?script=sci_arttext&pid=S0011-85162018000300006&lng=en

Author’s Response:

1) Thank you for your comment.

We appreciate your observation regarding our interpretation of Reference 42. Upon review, we agree that the Hayes and Ingram study primarily highlights the lack of awareness and clarity regarding academic career pathways among dental clinicians in Australia. The study involved 85 dental clinicians, of whom 79% reported that there was no clear pathway to an academic career and this lack of a clear career pathway was perceived as a barrier to academic career. We have revised our manuscript to more accurately reflect this focus on perceived barriers and limited awareness and we have also included Mostert (2018) as a valuable comparative perspective.

The new paragraph with updated citations (page 23):

“Dental academics are required to manage a wide range of responsibilities including teaching, research, clinical duties, and administration which contributes to a demanding work environment. This complexity can lead to challenges such as academic identity crisis [42], reduced job satisfaction, and hindered career progression. Regardless of the factors that motivate clinicians to pursue academic careers, whether it is limited awareness and unclear pathways that may discourage entry [43], or the appeal of intellectual engagement and job security that may encourage it [44], defining the core roles of dental academics remains essential. Clear role definitions are important to support academic identity development, guide career progression, and inform institutional policies across diverse educational settings. Furthermore, the limited literature outlining faculty competencies and role expectations in medical and dental education [45] highlights a significant gap that this study seeks to address.”

The study design is appropriate, and has academic merit as it attempts to address an important topic that could address issues in dental education requiring some extent of committed and support to dental academics among key decision makers in dental education. We thank the reviewer for recognising the academic merit of our study and its relevance to current challenges in dental education. We agree that addressing the multifaceted roles of dental academics is vital for informing institutional planning, resource allocation, and professional development strategies. It is our hope that the findings from this study will support efforts to guide decision-makers in better understanding the role-specific needs of dental academics and the importance of structured support systems for their career progression.

Reviewer 1:

2) Details of methods and analysis applied in this study can be improved. It is not clear how the experts panellists where identified?

Author’s Response:

2) We appreciate this comment. We have revised the manuscript to more clearly describe how experts were purposively selected based on specific inclusion criteria, including a minimum of five years of academic experience and active involvement in multiple dental academic roles. To preserve participant anonymity, we have not listed individual names but have included a summary table outlining their country of origin, professional roles, and years of experience. This additional detail is now presented in the revised Methods section and in Table 4.

The new updated paragraph (page 9) :

“Based on the inclusion and exclusion criteria (Table 3), a purposive sampling strategy was used to identify potential experts from three countries. Expert panellists were identified through institutional websites and publicly available curricula vitae, with selection based on their active involvement in multiple academic roles such as teaching, research, clinical practice, and administration

Invitations were sent via email to 37 identified experts from 13 dental institutions and healthcare facilities, and those who consented to participate were subsequently provided with the Fuzzy Delphi Method (FDM) questionnaires through the SurveyMonkey platform. A summary of the demographic and professional characteristics of the expert panel is presented in Table 4.”

The results of the pilot study for the Fuzzy Delphi Method was not analysed and published. This results should have been published or presented as it would have provided valuable insights into the effectiveness of the questionnaire and justification for refining before conducting first round of FDM. This could also assist those wishing to repeat similar study. Result of the pilot study could contribute to the understanding of FDM application and potential challenges. We appreciate the reviewer’s comment regarding the importance of presenting the pilot study findings. We would like to clarify that the results of the pilot study have been included in the manuscript under the "Fuzzy Delphi Method Questionnaire Reliability" subheading within the Results section.

In this section, we presented the reliability analysis of the initial questionnaire using Cronbach’s alpha, which demonstrated a high level of internal consistency. Additionally, we reported on how feedback from the pilot participants was used to refine the wording and structure of several items before launching the first round of the Fuzzy Delphi Method. This process ensured content clarity and improved instrument validity, which we agree is crucial for those seeking to replicate similar studies.

Nonetheless, based on the reviewer’s suggestion, we have reviewed this section and have made minor revisions to enhance its visibility and clarify the linkage between the pilot testing and subsequent questionnaire refinement.

In the updated method section (page 9) :

“The questionnaire was pre-tested and subsequently piloted with 12 dental academics who were not part of the main expert panel. This pilot study was conducted prior to the first round of the Fuzzy Delphi Method to assess the reliability, clarity, and relevance of the questionnaire items. Amendments were made based on pre-test feedback to improve wording and structure. During the pilot phase, the internal consistency of the constructs was evaluated using Cronbach’s alpha (α), standard deviation (SD), inter-item correlation (r), and corrected item-total correlation (CITC). Findings from the pilot test were used to refine item phrasing and ensure reliability before conducting the first round of the FDM. These results are reported under the 'Fuzzy Delphi Method Questionnaire Reliability' subheading.

In the updated results section (page 14) :

“Based on the reliability analysis conducted during the pilot study, all four main constructs demonstrated acceptable to good internal consistency. The major element “definition as a clinician” achieved an alpha (α) of 0.760, while “definition as a researcher” attained an alpha (α) of 0.775, both indicating acceptable reliability values. Similarly, “definition as an administrator” scored an alpha (α) of 0.806, and “definition as a teacher” scored an alpha (α) of 0.808, demonstrating a good level of reliability (Table 6). Given these findings, the questionnaire was deemed reliable and was retained without major modifications for use in the subsequent Delphi rounds. The finalised questionnaire consisted of 16 items representing various role definitions of dental academics.

Reviewer 1:

3) There are no clearly defined objectives; the Fuzzy Delphi method is designed to gather expert consensus on a specific topic, so the research needs to have a clear goal in mind to guide the selection of experts, the questions asked, and the interpretation of results. These objectives should be aligned with the current knowledge and concerns within the field of study.

Author's response:

3) Thank you for your insightful comment. We have revised the manuscript to clearly state that this study aims to systematically define the specific roles of dental academics using the Fuzzy Delphi Method (FDM). This represents the first part of a larger research project that investigates the attributes and competencies required for each of the identified dental academic roles. By establishing role definitions as a foundation, this study supports subsequent phases that will explore the skills and competencies necessary for career development and professional growth in dental academia.

The new updated paragraph (pages 4-5) :

“This study represents the first phase of a larger research project aimed at developing a comprehensive Dental Academic Career Pathway (DACP) model. It seeks to systematically define the specific roles of dental academics using the Fuzzy Delphi Method (FDM), forming a foundational step toward understanding the attributes, competencies, and developmental needs within dental academia. Through expert consensus, this study offers essential role definitions that can support academic identity formation and inform institutional policy and professional development. While the questionnaire was constructed through literature mapping and expert input, the broader study adopts the Social Cognitive Career Theory (SCCT) as its conceptual lens. SCCT will be applied in later phases to interpret how clearly defined academic roles may influence self-efficacy, outcome expectations, and motivation to pursue or sustain a career in dental academia.”

There are a number of issues with the methods and analysis that need to be clarified/addressed; furthermore, some of the conclusions overreach the data collected, while other important results are given less emphasis that they warrant. How where the experts identified, and how was their annonimity maintained. It is not clear how the individual expert opinions were aggregated and adjusted towards a consensus. This should be reported. Thank you for your feedback. We would like to clarify that this study employed the Fuzzy Delphi Method (FDM), in which each expert panellist was contacted individually and provided their responses independently via the Survey Monkey platform. There was no interaction or discussion among the experts, ensuring anonymity and reducing the risk of groupthink or dominant influence.

Consensus was not reached through group discussion but was instead determined quantitatively using the established FDM criteria. Specifically, an item was accepted only when all three of the following conditions were fulfilled:

1. The threshold (d) value was ≤ 0.2,

2. The group consensus percentage exceeded 75%, and

3. The fuzzy score (α-cut value) was > 0.5.

This rigorous process ensured objectivity and consistency in determining consensus. We have revised the relevant sections of the manuscript to more clearly explain these methodological details.

The new updated paragraph (page 13) :

“The consensus process in this study was conducted entirely through a quantitative approach. Each item in the questionnaire was analysed using the Fuzzy Delphi Method (FDM), and items were retained only if they met all three standard conditions: (1) the threshold value (d) reflecting the distance between experts’ opinions was ≤ 0.2; (2) the percentage of agreement among panellists exceeded 75%; and (3) the fuzzy score (α-cut value) was greater than 0.5. These thresholds are consistent with established FDM practices and were applied to ensure that only items with strong consensus and perceived relevance were included. This rigorous and structured approach reduces subjectivity and enhances the reliability and validity of the findings.”

It is overall a valuable study, in institutions where the different roles are not clearly defined. We sincerely thank the reviewer for recognising the value of this study. Indeed, this was a key motivation behind our work, to systematically define the multifaceted roles of dental academics and address the ambiguity often observed in academic settings. By establishing clearer role definitions, we hope to support faculty development, improve institutional clarity, and contribute to the broader discourse on academic identity and progression in dental education.

Reviewer 2:

Paper is well written and structure. My quires and suggestions are given as follows:

1) What exact criteria or decision rules were used in selecting and modifying the items between FDM rounds? Please provide examples of how feedback was used to generate the 9 new items in Round 2.

Author's response:

1) We appreciate the reviewer’s request for greater clarity on the decision rules used between FDM rounds.

In the first round of the Fuzzy Delphi Method (FDM), each item was quantitatively evaluated against three core criteria:

1. The threshold value (d) ≤ 0.2

2. Expert consensus > 75%

3. Fuzzy score (α-cut value) > 0.5

Items failing to meet one or more of these criteria were excluded from Round 2.

In addition to the statistical analysis, qualitative feedback from expert panellists was critically reviewed. Experts were encouraged to provide open-ended comments, suggestions, and critiques on each role definition. This qualitative input was thematically analysed and used to guide refinement of existing items and development of new items.

For example, under the role of “Teacher,” Round 1 included the following four items:

1. An educator whose main tasks are educating, guiding, mentoring, teaching, directing, training, assessing, and evaluating students on the knowledge of dentistry and other related sciences.

2. A professional with a bachelor's/master’s qualification in dentistry and a mandatory basic training in teaching.

3. An individual who is the expert of the field being taught in the academic settings.

4. An individual who is trained and qualified with Bachelor’s or Master’s degrees in the education field.

Experts commented that these items did not sufficiently reflect clinical expertise, pedagogical training, or evidence-based teaching practices. In response, these items were revised for clarity and relevance, and new items were added. For instance:

• Item 2 was refined to:

"An individual who is the expert of the field being taught in the academic settings and practices evidence-based dentistry, and evidence-based education."

• Item 5 (new addition) was created based on expert input emphasising the need for practical clinical teaching:

"An individual who has adequate clinical experience and is able to guide the students using teaching methods that are updated and relevant."

These modifications reflect both the statistical outcomes and the expert recommendations and were intended to increase precision, relevance, and comprehensiveness of the constructs.

2) Can the authors clarify how subjectivity in assigning fuzzy numbers was minimized? Was there any calibration or standardization step among raters?

Author's response:

2) Thank you for the valuable question. In this study, subjectivity in assigning fuzzy numbers was minimised by using a standardised and literature-supported conversion mapping between Likert scale responses and Triangular Fuzzy Numbers (TFNs).

Experts were only required to respond using a 5-point Likert scale, and they did not assign any fuzzy values themselves, which removed individual variability in fuzzy interpretation.

The conversion used was presented in Table 5:

Likert scale Linguistic Variables Fuzzy Scales

1 Strongly Disagree 0.0 0.0 0.2

2 Disagree

---

## [Decision Letter · Decision Letter 1]

5 Aug 2025

Defining the Multifaceted Roles of Dental Academics: A Consensus Approach Using the Fuzzy Delphi Method

PONE-D-24-57785R1

Dear Dr. Ahmad,

We’re pleased to inform you that your manuscript has been judged scientifically suitable for publication and will be formally accepted for publication once it meets all outstanding technical requirements.

Kind regards,

Aamir Ijaz, MD, FCPS, FRCP, MCPS-HPE

Academic Editor

PLOS ONE

Additional Editor Comments (optional):

Reviewers' comments:

Reviewer's Responses to Questions

**Comments to the Author**

Reviewer #2: All comments have been addressed

2. Is the manuscript technically sound, and do the data support the conclusions?

Reviewer #2: Yes

3. Has the statistical analysis been performed appropriately and rigorously?

Reviewer #2: No

4. Have the authors made all data underlying the findings in their manuscript fully available?

Reviewer #2: Yes

5. Is the manuscript presented in an intelligible fashion and written in standard English?

Reviewer #2: Yes

Reviewer #2: Paper is revised and it can be accepted now in it's current form. Thank you for addressing my comments

**Do you want your identity to be public for this peer review?** For information about this choice, including consent withdrawal, please see our Privacy Policy

Reviewer #2: **Yes: ** Arvind Kumar Prajapati

---

## [Editor Report · Acceptance letter]

PONE-D-24-57785R1

PLOS ONE

Dear Dr. Ahmad,

I'm pleased to inform you that your manuscript has been deemed suitable for publication in PLOS ONE. Congratulations! Your manuscript is now being handed over to our production team.

Kind regards,

on behalf of

Professor Aamir Ijaz

Academic Editor

PLOS ONE